# Diatom-Inspired Structural Adaptation According to Mode Shapes: A Study on 3D Structures and Software Tools

**DOI:** 10.3390/biomimetics9040241

**Published:** 2024-04-18

**Authors:** Simone Andresen, Ahmad Burhani Ahmad Basri

**Affiliations:** Alfred Wegener Institute Helmholtz Centre for Polar and Marine Research, Am Handelshafen 12, 27570 Bremerhaven, Germany; ahmad.basri@awi.de

**Keywords:** biomimetics, coding, C#, diatom-inspired, eigenfrequency maximization, finite element analysis, lightweight design, Synera software

## Abstract

Diatoms captivate both biologists and engineers with their remarkable mechanical properties and lightweight design principles inherent in their shells. Recent studies have indicated that diatom frustules possess optimized shapes that align with vibrational modes, suggesting an inherent adaptation to vibratory loads. The mode shape adaptation method is known to significantly alter eigenfrequencies of 1D and 2D structures to prevent undesired vibration amplitudes. Leveraging this insight, the diatom-inspired approach to deform structures according to mode shapes was extended to different complex 3D structures, demonstrating a significant enhancement in eigenfrequencies with distinct mode shapes. Through extensive parameter studies, frequency increases exceeding 200% were obtained, showcasing the method’s effectiveness. In the second study part, the studied method was integrated into a user-friendly, low-code software facilitating swift and automated structural adjustments for eigenfrequency optimization. The created software tools, encompassing various components, were successfully tested on the example structures demonstrating the versatility and practicality of implementing biomimetic strategies in engineering designs. Thus, the present investigation does not only highlight the noteworthiness of the structural adaptation method inspired by diatoms in maximizing eigenfrequencies, but also originate software tools permitting different users to easily apply the method to distinct structures that have to be optimized, e.g., lightweight structures in the mobility or aerospace industry that are susceptible toward vibrations.

## 1. Introduction

Diatoms (Figure 1), unicellular organisms ubiquitous in marine and freshwater environments, have long captivated scientists and engineers alike with their intricate silica cell walls, known as frustules. These microscopic structures, comprised of two valves joined by silica bands [1,2], exhibit remarkable diversity in shape and size across the estimated 20,000–200,000 recent diatom species [3,4]. Ranging from circular to oval, stick-shaped, or star-like configurations, with dimensions spanning from micrometers to one millimeter, diatom frustules represent a great example of nature’s ingenuity in biological design [1,2].

As diatoms need to float in the nutritious upper water column to survive and to carry out photosynthesis, their silicate shells must be very light. At the same time, high structural stiffness and stability are essential for withstanding the attacks of predators such as copepods. Thus, the combination of low mass and high resilience leads to lightweight design principles that can be observed in these biological structures [6,7,8,9].

Although the highly complex frustules show the same basic construction principles of technical lightweight structures using ribs, lattices, and honeycombs [9], they embody high multifunctionality with attributes including high permeability [10], mechanical robustness [11], high energy absorption [7], and vibration optimization [12]. This versatility positions diatom shells as promising candidates for a diverse array of technical applications, from enhanced nano-filtration [13], to drug delivery systems [14], plastic deformation elements in automobiles [15], and optical sensing materials [1,16]. The exceptional mechanical properties and lightweight design principles inherent in diatom shells have inspired successful integration into structures across industrial sectors, evidencing their practical utility and potential for advancing lightweight construction methodologies [5,17,18,19].

Aside from the lightweight design principles observed in diatom shells, a high impact of these biological structures on the vibration characteristics is expected, since copepods also hammer on their prey to crack it [20]. Consequently, diatom frustules must exhibit resilience against vibratory loads. However, the literature on the vibration characteristics of diatom shells remains limited, with only a few published studies addressing this topic [12,21,22,23]. Notably, one such study [12] conducted numerical modal analyses on abstracted diatom shells, revealing a strong correlation between vibrational mode shapes and observed deformation patterns. This correlation suggests that diatom frustules may be optimized for vibratory loads. Employing a biomimetic approach, inspired by the ’biology-push’ methodology [24], the observed mode shape adaptation in diatoms is further investigated and applied to technical structures to enhance the vibration characteristics and mitigate undesired vibrations. This approach aims to leverage the insights from biological systems to inform and improve engineering designs, thereby enhancing the structural performance.

In applied engineering contexts, unwanted vibrations present significant challenges, potentially leading to functional impairment or structural failure, particularly in lightweight constructions that are usually very prone to vibrate or flutter [25,26]. Traditional methods to avoid high vibration amplitudes include the implementation of damping devices or an increase in structural mass. However, both methods would increase the structural mass and thus conflict with the overarching goal of a lightweight design [27]. In addition, active damping devices often used in lightweight structures also increase the complexity of the entire system.

In regard to solutions for undesired vibrations in lightweight structures, the selection of materials with superior damping properties can prevent high vibration amplitudes. However, high damping materials often show a reduced stiffness. Fiber-reinforced polymer composites, though, can be stiff, lightweight structures with high damping properties depending on the choice of matrix and fiber [28]. However, many geometries cannot be made out of these composites. Worth mentioning are also lightweight advanced materials (e.g., mechanical metamaterials with ultra-high performance) for damping insulation, but these materials are comparably expensive, difficult to implement, and manufacturing techniques are not yet well-established [29]. Recently, particle damping has been utilized in order to prevent high vibration amplitudes in lightweight structures, as it only leads to a comparably little mass increase. Yet, the prediction of damping parameters is difficult and trial-and-error experiments are still common procedures [30]. Aside from this, sandwich structures–if applicable–can be stiff and lightweight components with high damping properties, if the material and structure of core and face sheets are ideally selected [26]. However, sandwich structures increase the complexity of the entire system because they usually are comprised of different materials and are often comparably complex to manufacture. Consequently, there is a pressing need for innovative, easy-applicable optimization strategies that can effectively address vibration concerns without compromising structural integrity, nor increasing the complexity of the system itself, nor adding mass.

Previous research efforts have explored various optimization approaches to mitigate vibrations [31,32,33,34], yet many suffer from limitations. Structural optimizations, while effective, are often laborious and computationally intensive [35], particularly when applied to three-dimensional structures. Moreover, the manufacturing of resultant structures, typically intricate and complex, necessitates substantial financial resources and extended production timelines, underscoring the ongoing demand for efficient optimization methods in this domain. The application of these methods to complex engineering systems remains underexplored, necessitating approaches that can offer efficient solutions. While in ‘traditional engineering approaches’, engineers develop and apply structural optimizations techniques that are usually directed towards a known target (optimization objective), biologically inspired approaches are often highly complex due to the interdisciplinarity (biologists, physicists, and engineers working together). However, the usually immense bio-inspired studies, which cover basic research on organisms, the creation of (mathematical) models to understand and describe the discovered biological function, and the implementation of the findings to technical products, allowing for the discovery of new ideas to solve problems in the engineering world.

Regarding approaches to mitigate undesired vibrations, one notable strategy involves increasing (maximizing) structural eigenfrequencies, thereby ’detuning’ the system and mitigating resonance phenomena caused by the alignment of a structure’s eigenfrequencies with external vibration frequencies. This proactive approach seeks to shift eigenfrequencies away from critical frequency values to prevent resonance, thereby minimizing vibration amplitudes. Drawing inspiration from observations in diatom shells, adapting a structure’s shape according to a mode shape presents a promising avenue for efficient eigenfrequency shifts. Previous studies [36,37] have demonstrated the efficacy of mode shape adaptation in significantly increasing the eigenfrequencies of one- (1D) and two-dimensional (2D) structures, respectively. In another study, mode shapes were applied as embossing pattern to a truck cab floor panel to shift the mode shape frequencies [38]. However, while three-dimensional (3D) structures are prevalent in technical applications, the application of mode shape adaptation to such structures remains largely unexplored. Additionally, given the complexity and time-consuming nature of simulating mode shape adaptation, the development of software capable of automating this process for structures of varying dimensions and shapes is crucial.

In summary, the present study focuses on the application of a structural adaptation method according to mode shapes, inspired by diatom frustules, to enhance the vibration characteristics of technical structures. The primary objectives of this research endeavor are (1) to evaluate the efficacy of the diatom-inspired optimization method in efficiently manipulating the eigenfrequencies of three-dimensional structures and (2) to integrate the structural adaptation method into a low-code software platform capable of automating the mode shape adaptation process across structures of varying dimensions, encompassing 1D, 2D, and 3D configurations. By addressing these objectives, this study seeks to contribute to the understanding and practical implementation of biologically inspired optimization techniques in engineering contexts.

## 2. Materials and Methods

Since this study investigates the impact of structural adaptation according to mode shapes on the eigenfrequencies, a brief introduction to mechanical vibrations focusing on the information related to this work is given.

Mechanical vibrations denote the oscillation of an elastic body due to disturbances that move the body away from its equilibrium position. All bodies with a mass and a finite stiffness are capable of vibrating. The following paragraph briefly summarizes information closely related to the present study. For more information it is referred to [39].

Taking a body off its equilibrium position leads to a free vibration, characterized by a characteristic eigenfrequency (also called ’natural frequency’) and a characteristic mode shape (also called ’eigenmode’ or ’eigenvector’). If the frequency exciting a body to vibrate corresponds to the body’s eigenfrequency, a large vibration amplitude occurs (resonance phenomenon). Each mode shape of a structure is associated with a specific eigenfrequency. Modal analyses, using, for example, numerical finite element (FE) analyses, permit the determination of eigenfrequencies and mode shapes. The mode shapes are ordered by ascending frequency values. Thus, if structural adaptation leads to eigenfrequency shifts, the mode shapes can switch order (’mode switching’). In regard to the depiction of mode shapes obtained from modal analyses, structural deformations are normalized to easily visualize areas of comparably high and low deformations within one structure.

### 2.1. Bio-Inspired Mode Shape Adaptation of 3D Structures

Different 3D structures were pre-deformed according to their 1st and 2nd mode shapes to study the impact of the pre-deformations on the eigenfrequencies and the mode shape frequencies. The models were constructed using the software Rhinoceros (version 6 SR9, Robert McNeel & Associates, Seattle, WA, USA) and its plug-in Grasshopper (version 1.0.0007, Robert McNeel & Associates, Seattle, WA, USA) as well as the tool ELISE (version 1.0.38, Synera GmbH, Bremen, Germany) running within Grasshopper. Numerical finite element analyses were conducted using the solver OptiStruct (Altair®HyperWorks®Version 2019, Altair Engineering, Inc., Troy, MI, USA). Within the Grasshopper environment, the entire analysis process was set up, including the procedure visible in Figure 2. After the construction of the 3D geometry, the finite element mesh was generated. The definition of the boundary condition and the modal load case led to the modal analysis, resulting in the structure’s eigenfrequencies and mode shapes. The eigenvectors were extracted and normalized by dividing each eigenvector by the highest absolute eigenvector component.

The normalized eigenvectors, showing values within [−1;1], were multiplied by a factor $\delta_{\max}$ representing the maximum pre-deformation. Divided by the structure’s wall thickness *t*, the maximum relative pre-deformation δ was obtained according to [36,37]: (1)$$\delta =\frac{\delta_{\max}}{t}$$

The mesh of the 3D structure was deformed multiplying the maximum pre-deformation $\delta_{\max}$ with each mesh node to form the mesh adapted according to a mode shape. Modal analysis was then performed again to obtain the eigenfrequencies and mode shapes of the pre-deformed 3D structure. The eigenfrequency and mode shape frequency deviation Δf, compared to those of the undeformed 3D structure, was obtained using the following equation calculating the deviation of a frequency $f_{b}$ compared to a frequency $f_{a}$: (2)$$\Delta f=\frac{f_{b}-f_{a}}{f_{a}}\;\cdot \;100$$

The mode shape adaptation method was applied to seven different 3D structures visible in Figure 3. The hollow cuboid, the thick hexagonal prism, and the cube (Figure 3a–c) were modeled with volume elements, while the truncated pyramid, the thin hexagonal prism, the curved rectangular duct, and the connector square (Figure 3d–g) had small wall thicknesses and were therefore modeled with shell elements. All surfaces and edges highlighted in red in Figure 3 were clamped, i.e., all translations and rotations were inhibited. The mesh convergence studies indicated the sufficient element sizes (Appendix A). Table 1 summarizes the mesh properties, material properties, applied maximum pre-deformations, and the mass.

Each structure was pre-deformed according to its 1st and 2nd mode shapes (Figure 4). Within a parameter study, the deformations were increased from values to large pre-deformations. For the three structures meshed with volume elements, the following maximum pre-deformations were studied:Hollow cuboid: 0 mm to 150 mm in steps of 15 mm;Thick hexagonal prism: 0 mm to 5 mm in steps of 1 mm, and 5 mm to 25 mm in steps of 5 mm;Cube: 0 mm to 5 mm in steps of 1 mm, and 5 mm to 30 mm in steps of 5 mm.

For the remaining four structures, the applied maximum pre-deformations are listed in Table 2. The constant wall thicknesses were slightly reduced for larger pre-deformation values to maintain the constant mass.

For all models, the first four eigenfrequencies, as well as the frequencies of the first four mode shapes, were recorded and the frequency deviations were obtained using Equation (Equation 2). However, due to the structural adaptation, mode shape orders changed in some cases. If a certain mode shape was no longer visible within the studied mode shapes, it was not recorded further.

### 2.2. Integration of the Bio-Inspired Adaptation Method into a Low-Code Software

#### 2.2.1. Automatization of the Mode Shape Adaptation Method

The preliminary outcomes of antecedent studies [36,37] and the study on 3D structures presented here have shown the great potential of utilizing the mode shape adaptation method in diverse engineering applications. Therefore, further exploration with an emphasis on automating the structural modification process based on the mode shape adaptation method was performed. The approach was transformed into algorithms seamlessly integrated into the low-code product development software Synera (version Inspiring Icarus (V10.0.17), Synera GmbH, Bremen, Germany). The algorithms function as modular tools accessible to software users, enabling rapid and automated structural adjustments to alter eigenfrequencies and/or augment the stiffness.

Figure 5 illustrates the automated process of the method implemented to generate new Synera components. The automated process simplifies the workflow presented in Figure 2, focusing only on pre-deforming the initial structure, i.e., on the method itself. Thus, software users can choose to set different boundary conditions or load cases before or after the model is pre-deformed. In this way, users will have more flexibility and freedom to conduct any type of subsequent analysis.

Three new Synera components were created based on a written C# code. Figure 6 shows the developed beam, shell, and solid components that pre-deform 1D, 2D, and 3D structures according to mode shapes. Each component was created for models meshed with certain type of elements, i.e., beam elements, shell elements, or volume elements. The inputs and the outputs of the components are explained in Figure 7.

Figure 8 exemplarily shows the Synera workflow for creating a pre-deformed beam model. This model was studied to validate the created component (cf., Section 2.2.2).

#### 2.2.2. Validation of the Created Software Components

The purpose of this section is to validate the results produced by the created software components compared to analytical calculations or other studies. Here, all parametric and algorithm-based constructions were performed using Synera, which allowed the construction of the entire design and simulation process and the connection to the numerical solver OptiStruct, as exemplarily shown in Figure 8.
(i)Mode Shape Adaptation of Beam Components

A beam with a circular cross-section was pre-deformed according to its 1st mode shape to investigate the functionality of the created software component (Figure 6a) by comparing the results with analytical calculations. Table 3 lists the properties of the beam. The beam was constrained at both ends except for the rotation around the *y* axis, i.e., perpendicular to the beam length, implying a simply supported boundary condition. A modal analysis to calculate the eigenfrequencies and mode shapes was conducted. The adequate mesh size was obtained in a mesh convergence study on the undeformed beam, which resulted in a total of 25 nodes and 24 CBEAM elements (see Appendix A). CBEAMs are beam elements with two nodes and six degrees of freedom per node that allow tension, compression, and bending. Figure 9 shows the undeformed ($\delta_{\max}$ = 0 mm) and exemplarily four pre-deformed beams. Maximum pre-deformations of 1 mm, 2 mm, 3 mm, 4 mm, 5 mm, 10 mm, 15 mm, and 20 mm were studied. For each simulated beam model, the frequency of the 1st and the 3rd mode shapes of the undeformed beam (i.e., the 1st and 2nd bending mode shapes in the xz plane) were recorded to investigate the frequency increase due to the mode shape adaptation method.

The obtained results were compared to analytical results using the Bernoulli beam theory. For the simply supported, undeformed beam characterized by a Young’s modulus *E*, material density ρ, length *l*, circular cross-section area *A*, and a second moment of inertia *I*, where both *A* and *I* depended on the beam diameter *d*, the *i*-th eigenfrequencies $f_{i,b}$ are defined as follows [40]: (3)$$f_{i,b}=\frac{1}{2\pi }\frac{{\left(i\pi \right)}^{2}}{l^{2}}\sqrt{\frac{EI}{A\rho }}\;\;\;\;;\;\;\;\;i=1,2,3,...$$
where $A=\pi d^{2}/4$ and $I=\pi d^{4}/64$. The *i*-th bending mode shape $\Phi_{i}$ of the beam in the xz plane can be described as
(4)$$\Phi_{i}\left(x\right)=\sin\left(\frac{i\pi x}{l}\right)\;\;\;\;;\;\;\;\;i=1,2,3,...$$

Formulae to calculate the 1st eigenfrequency of a circular arch have been published in [41] and were implemented in this study as an approximation for sinusoidal beams with small ratios of l⁄p, where *p* is the radius of the arch curvature and *l* is the undeformed beam length (see Figure 10). The frequency of the 1st mode shape $f_{M1,b}$ and the 3rd mode shape $f_{M3,b}$ of the beam pre-deformed according to the 1st mode shape were calculated as
(5)$$f_{M1,b}=\frac{1}{2\pi }\sqrt{\frac{1}{p^{4}}\left[0.82{\left(\frac{p}{k}\right)}^{2}+{\left(1-\frac{\pi^{2}}{\alpha^{2}}\right)}^{2}\right]\frac{EI}{A\rho }}$$
(6)$$f_{M3,b}=\frac{1}{2\pi }\sqrt{\frac{1}{p^{4}\alpha^{4}}\left[\frac{\alpha^{4}-8\pi^{2}\alpha^{2}+16\pi^{4}}{1+0.075\alpha^{2}/\pi^{2}}\right]\frac{EI}{A\rho }}$$
where α is the central angle (in radian), and *k* is the slenderness ratio of the circular cross-section. *p*, α, and *k* are given as
(7)$$p=\frac{1}{2h}\left[{\left(\frac{l}{2}\right)}^{2}+h^{2}\right]$$
(8)$$\alpha =2\sin^{-1}\left(\frac{l}{2p}\right)$$
(9)$$k=\frac{I}{A}=\frac{r^{2}}{4}$$

*h* is the arc height, which is equivalent to the maximum pre-deformation $\delta_{\max}$, and *r* is the cross-section radius.

The deviation between the mode shape frequencies obtained by the beam component and the analytically obtained values was calculated using Equation (Equation 2).
(ii)Mode Shape Adaptation of Shell Components

The investigation involved the application of the created software component (Figure 6b) to a square plate structure studied in [37]. In the following, the 2D structure is briefly described.

The considered square plate was characterized by an edge length of 100 mm and a constant thickness *t* of 2 mm (Figure 11). The material properties were the same as for the beam (Table 3). The plate model was meshed with 10,000 CQUAD4 elements as in the published study [37]. CQUAD4s are quadrilateral elements with four nodes and six degrees of freedom per node, suitable for simulating membrane or shell components. Concerning the boundary conditions, all degrees of freedom of the edge nodes were restricted (clamped). The plate model was pre-deformed according to its 1st mode shape using the created software component (Figure 2b). The studied maximum pre-deformations ranged from 1 mm to 40 mm. In accordance with [37], the thickness of the pre-deformed plates was adapted to maintain a constant mass. For each model, a modal analysis was performed in order to obtain the first six eigenfrequencies. The results were then compared to those presented in [37]. The average percentage of eigenfrequency deviation $\Delta f_{\mathrm{compa}}$ (from $f_{1}$ to $f_{6}$) was calculated using the following equation: (10)$$\Delta f_{\mathrm{compa}}=\frac{\sum_{i=1}^{n}\left|\frac{f_{\mathrm{created}\;\mathrm{component},i}-f_{\mathrm{reference},i}}{f_{\mathrm{reference},i}}\;\cdot \;100\right|}{n}\;\;\;\;;\;\;\;\;i=1,2,3,...,n$$
where *n* is the number of modes of interest.

(iii)Mode Shape Adaptation of Solid Components

In order to validate the accuracy of the developed software component for volume meshes (Figure 6c), the hollow cuboid meshed with volume elements was revisited. A pre-deformed structure is exemplarily illustrated in Figure 12a. Here, the results of the hollow cuboid obtained from Section 2.1 were compared to those obtained via the created component using Equation (Equation 10).

## 3. Results

### 3.1. Bio-Inspired Mode Shape Adaptation of 3D Structures

The mode shape adaptation approach applied to different 3D structures (see pre-deformed structures in Figure 12) increased the eigenfrequencies from $f_{1}$ to $f_{4}$ and mode shape frequencies from $f_{M1}$ to $f_{M4}$ with rising maximum relative pre-deformation for almost all studied models (Figure 13, Figure 14, Figure 15, Figure 16, Figure 17, Figure 18 and Figure 19). The results for the mode shape adaptation according to mode 2 are attached in the Appendix A. For the truncated pyramid (Figure 16), the thin hexagonal prism (Figure 17), the curved rectangular duct (Figure 18), and the connector square (Figure 19), the studied frequencies first increased strongly. However, with increasing maximum relative pre-deformation, the slope decreased. The frequencies of the hollow cuboid (Figure 13) increased with a nearly constant slope for all studied pre-deformations, while the thick hexagonal prism (Figure 14) initially showed frequency decreases before the frequencies then increased again for high maximum relative pre-deformations. The frequencies of the cube (Figure 15) remained almost constant through all studied pre-deformations.

The recorded mode shapes were tracked to discover the frequency alteration of each mode shape. The frequencies of the first mode shapes were almost completely tracked for the hollow cuboid, the truncated pyramid, and the thin hexagonal prism. Mode shape switches were partly visible, i.e., the mode shape order changed due to the structural deformations (e.g., Figure 13b and Figure 16b). However, the mode shapes themselves (i.e., the deformation) of the curved rectangular duct and the connector square varied strongly with the increasing maximum relative pre-deformation, so that the mode tracking could only be partly performed. For the thick hexagonal prism, the mode shape tracking did not succeed. Especially in the case of the hollow cuboid and the truncated pyramid, the mode shape frequencies increased more strongly than the eigenfrequencies, as the structural adaptation resulted in mode shape switches.

Generally, the highest frequency increases were obtained for the connector square, the thin hexagonal prism, and the truncated pyramid. A pre-deformation according to the first mode shape increased the first mode shape frequency by almost 300% (δ = 13.3) for the connector square, 165% (δ = 10.5) for the thin hexagonal prism, and 134% (δ = 13.9) for the truncated pyramid. The rectangular duct and the hollow cuboid showed first mode shape frequency increases of 47% (δ = 7.6) and 40% (δ = 15.0), respectively, for pre-deformations according to the first mode shape. However, for the thick hexagonal prism and the cube, the frequencies were hardly affected by the mode shape adaptation method. Similar results were obtained for the structural adaptation according to the second mode shape. Notably, while the frequency of the second mode shape rose up to 243% (δ = 13.3) for the connector square, the second highest increase was 164% (δ = 15.0) for the hollow cuboid. The second mode shape frequency of the curved rectangular duct was only slightly increased. Though it may be noted that the mode shape tracking was only partly possible, and high order mode shapes were not studied.

Finally, relating the obtained frequency increases to the structure’s mode shapes (Figure 4) indicates that structures with local/discrete mode shapes (i.e., clearly visible buckles) like the hollow cuboid, the curved rectangular duct, and the connector square are likely to show significantly raised frequencies. In contrast, the frequencies of structures showing mode shapes characterized by global deformations, e.g., the cube, are apparently hardly affected by the mode shape adaptation method. This phenomenon is clearly visible comparing the thick hexagonal prism to the thin hexagonal prism, where the latter exhibits discrete mode shapes and thus high frequency increases owing to the studied method.

### 3.2. Integration of the Bio-Inspired Adaptation Method into a Low-Code Software

The results of testing the created software components on a beam, shell, and solid structure are presented below.
(i)Mode Shape Adaptation of Beam Components

The outcomes derived from analytical calculations and the numerical computations employing the software components are depicted in Figure 20, while the frequency of the first mode shape shows a significant enormous increase, the frequency of the third mode shape remains almost constant for all studied pre-deformations. The numerically and analytically obtained frequencies exhibit a very high conformity for both studied mode shapes. The percentage of deviation of the frequency of the first mode shape was the highest at δ = 10 with 3.4%.

(ii)Mode Shape Adaptation of Shell Components

Regarding the study on the shell component, Figure 21 illustrates the outcomes of pre-deforming the square plate according to the first mode shape. Initially, a consistent increase in all frequencies was observed, followed by a subsequent decrease after reaching a maximum relative pre-deformation of δ = 17.4.

Upon comparing the results obtained from [37] and the developed software component, it is noteworthy that the correspondence is very precise. The maximum average percentage of difference across all pre-deformed models was less than 8.7%, as detailed in Table 4.
(iii)Mode Shape Adaptation of Solid Components

A comprehensive comparison was conducted between the results obtained from the created solid components and those from the preceding investigation outlined in Section 3.1 (cf., Figure 13). The maximum average percentage of deviation for all studied eigenfrequencies across all pre-deformed models amounted to less than 1.4% (Table 5).

## 4. Discussion

### 4.1. Bio-Inspired Mode Shape Adaptation of 3D Structures

The mode shape adaptation method was successfully applied to different 3D structures. Notably, certain structures exhibited considerable eigenfrequency increases following pre-deformation, whereas others displayed minimal changes in frequency. The analysis elucidated a correlation between mode shape characteristics and frequency alterations. Structures manifesting local deformations (i.e., clearly visible buckles), such as the hollow cuboid or connector square, experienced significant frequency increases. In contrast, those showing global deformations, like the cube, had their frequencies hardly changed. A direct comparison of the thin and the thick hexagonal prisms demonstrates this phenomenon clearly as follows: the thin structure exhibits pronounced frequency shifts due to localized buckles, while the thick prism showed marginal frequency impact due to global buckling patterns.

Previous research on 1D structures [36,37], and also the results within this paper, corroborated the feasibility of selectively increasing single mode shape frequencies while maintaining constancy in others. For simple 2D structures, already, the mode shape adaptation method significantly altered frequency [37]. Notably, the mode shapes remained traceable. The current investigation on different 3D structures revealed profound mode shape modifications, rendering mode tracking challenging, particularly under high pre-deformation conditions. It is expected that these pre-deformations could potentially shift mode shapes to unexplored frequency ranges beyond the scope of this study, necessitating a further analysis of higher-order mode shapes and frequencies to validate this assumption.

Summing up, the findings underscore the efficacy of the diatom-inspired mode shape adaptation method in strongly manipulating eigenfrequencies, extending its applicability beyond 1D and 2D structures to encompass complex 3D structures with distinctive mode shapes.

### 4.2. Integration of the Bio-Inspired Adaptation Method into a Low-Code Software

The present discussion centers around the outcomes obtained through analytical and numerical computations involving mode shape adaptation of beam, shell, and solid components, as presented in Section 3.1. The findings revealed a highly favorable correlation between the results. This observed correlation is indicative of the robustness and consistency of the created mode shape adaptation components. The alignment between the analytical and numerical results is a critical aspect, as it underscores the component’s ability to accurately capture and represent the dynamic behaviour of the system under consideration.

One of the pivotal aspects addressed in this study pertains to the accuracy of the mode shape adaptation components in predicting natural frequencies. The results indicate a commendable accuracy, with a very low average discrepancy between the compared results. The level of agreement between the manual numerical computations and the automated numerical computations via components created not only validates the modeling approach but also instills confidence in the capability of the components to fulfill their intended purpose effectively within the context of structural analysis.

Additionally, the uniformity of the solver employed across all studies conducted in this investigation merits consideration. This particular aspect assumes heightened significance, particularly when the objective is to attain consistency in results throughout a study. Notably, both comparisons conducted within the exemplarily application of the created shell and solid components were executed using the identical solver (OptiStruct). This deliberate choice emphasizes the commitment to precision and reliability, ultimately contributing to the attainment of a commendable level of accuracy in the presented results.

Apart from the accuracy and efficiency of the components, an additional noteworthy point to emphasize is the inclusive nature of the proposed components, which effectively democratizes their utilization. These components are designed to be accessible to a diverse range of users, irrespective of their background or expertise. Remarkably, users are not required to possess specialized knowledge, such as a background in biology, nor do they need a comprehensive understanding of structural vibrations. The user-friendly nature of these components ensures that individuals with varying levels of expertise can seamlessly employ the method. Furthermore, the components boast an innovative feature described as automated and smart structural adaptation inspired by diatoms, signifying a sophisticated and intuitive approach that facilitates adaptability without necessitating an in-depth comprehension of intricate structural dynamics.

Summing up, the diatom-inspired mode shape adaptation method has been successfully implemented into a user-friendly software, which permits an easy and efficient application of the studied method to diverse 1D, 2D, and 3D structures of different fields of application. In continuation studies, other results of research studies on diatom-inspired structures are also planned to be implemented in software tools to be available for users across different industry sectors.

## 5. Conclusions

The successful application of the mode shape adaptation method to various 3D structures highlights its efficacy in manipulating eigenfrequencies, with notable variations observed across different structural forms. The analyses revealed a correlation between the mode shape characteristics and frequency alterations, particularly evident in structures exhibiting local deformations, which experienced significant frequency increases compared to those showing global deformations. This phenomenon, previously observed in 1D and 2D structures, underscores the method’s ability to increase the eigenfrequencies of 3D structures.

Apart from that, through analytical and numerical computations across beam, shell, and solid components, the study demonstrates the robustness and consistency of the diatom-inspired structural adaptation method, with results showcasing a highly favorable correlation and commendable accuracy in predicting natural frequencies. The innovative automated and smart adaptation features offer a sophisticated yet intuitive approach, facilitating structural optimization to maximize frequencies without the need for an in-depth understanding of complex dynamics and enabling users of diverse backgrounds to employ them seamlessly without requiring specialized knowledge.

## Figures and Tables

**Figure 1 biomimetics-09-00241-f001:**
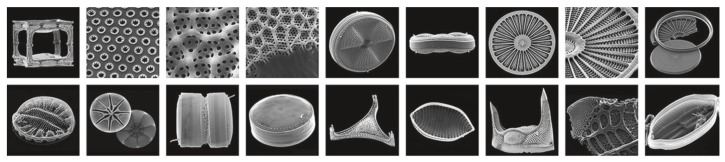
Scanning electron microscopic images of diatom frustules (adapted according to [5]).

**Figure 2 biomimetics-09-00241-f002:**
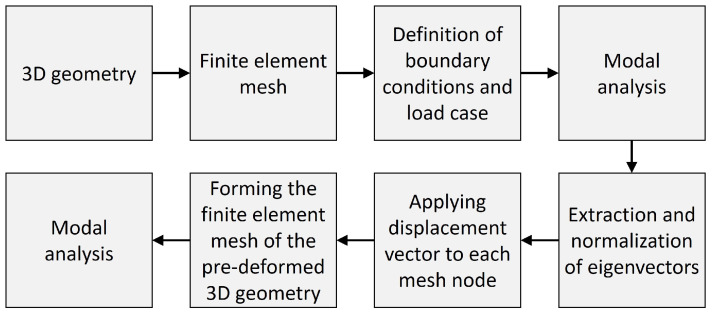
Procedure of the investigation.

**Figure 3 biomimetics-09-00241-f003:**
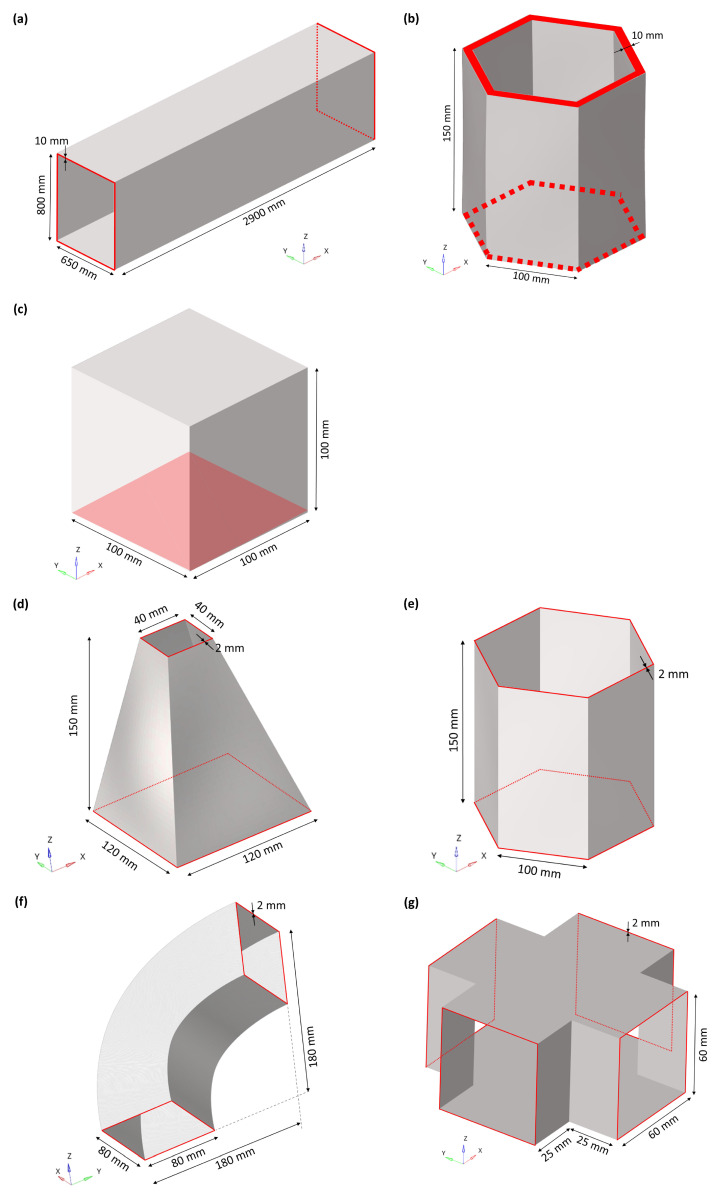
Three dimensional view of (**a**) hollow cuboid, (**b**) thick hexagonal prism, (**c**) cube, (**d**) truncated pyramid, (**e**) thin hexagonal prism, (**f**) curved rectangular duct, and (**g**) connector square investigated in the present study. The edges and surfaces highlighted in red indicate the defined clamped condition.

**Figure 4 biomimetics-09-00241-f004:**
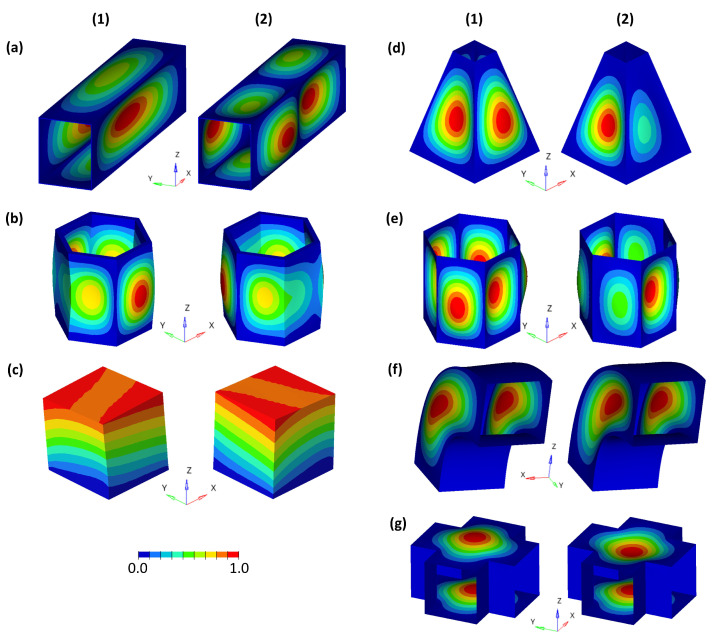
1st (1) and 2nd (2) mode shape of the studied structures (**a**) hollow cuboid, (**b**) thick hexagonal prism, (**c**) cube, (**d**) truncated pyramid, (**e**) thin hexagonal prism, (**f**) curved rectangular duct, and (**g**) connector square. The coloring represents the normalized vibration amplitude.

**Figure 5 biomimetics-09-00241-f005:**
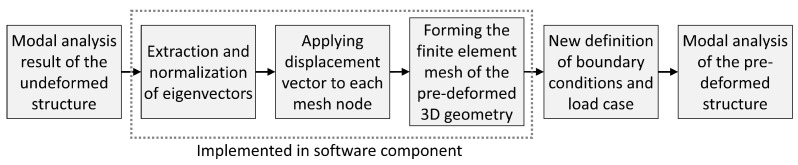
Automatization process of the bio-inspired mode shape adaptation method.

**Figure 6 biomimetics-09-00241-f006:**
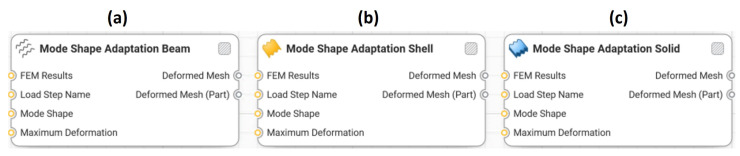
Created mode shape adaptation components for a (**a**) beam, (**b**) shell, and (**c**) solid model.

**Figure 7 biomimetics-09-00241-f007:**
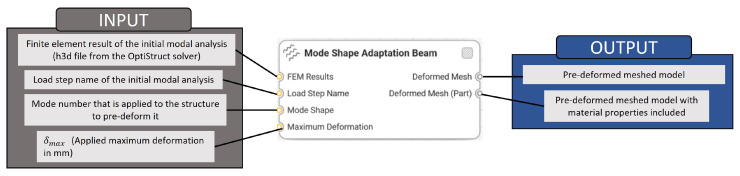
Explanation of the inputs and outputs of the created software components.

**Figure 8 biomimetics-09-00241-f008:**
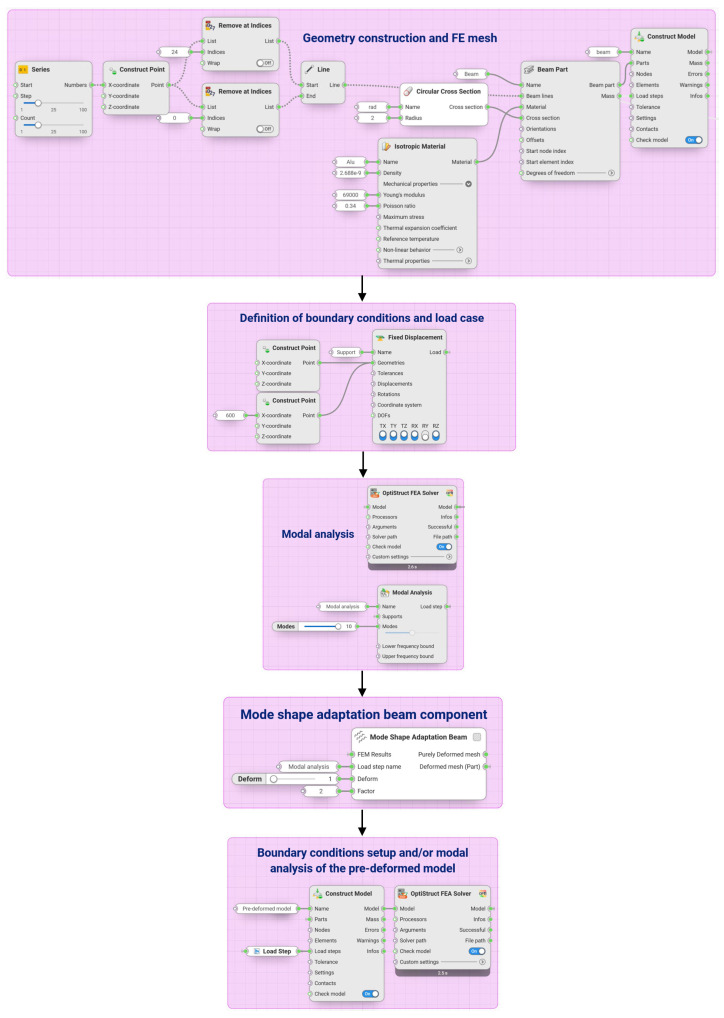
Example of a pre-deformed model creation process with in the Synera environment using the developed software component for beam models.

**Figure 9 biomimetics-09-00241-f009:**
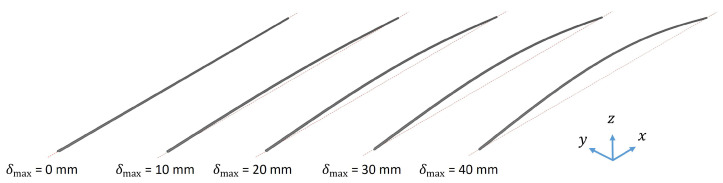
Three dimensional view of the undeformed beam ($\delta_{\max}$ = 0 mm) and the beam pre-deformed according to the 1st mode shape considering different maximum pre-deformations $\delta_{\max}$. The beam itself is colored in dark grey and the undeformed beam is illustrated with a red dashed line.

**Figure 10 biomimetics-09-00241-f010:**
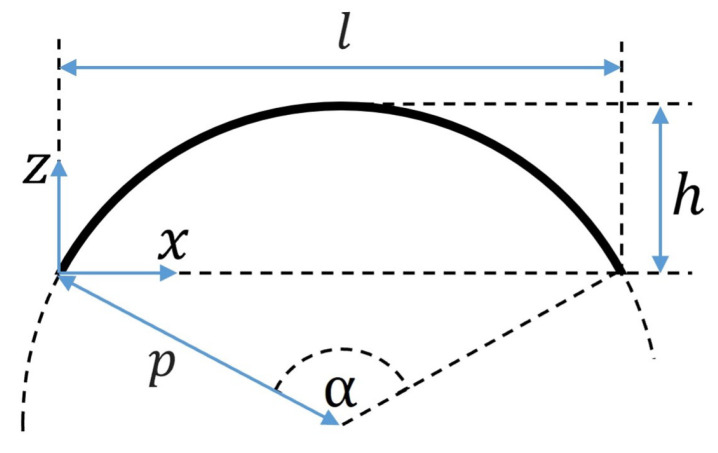
Circular arc dimensions.

**Figure 11 biomimetics-09-00241-f011:**
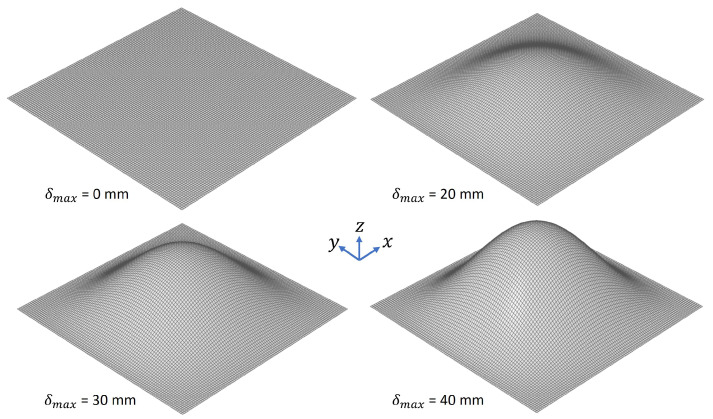
Three dimensional view of the undeformed plate ($\delta_{\max}$ = 0 mm) and the plate pre-deformed according to the 1st mode shape considering different maximum pre-deformations $\delta_{\max}$.

**Figure 12 biomimetics-09-00241-f012:**
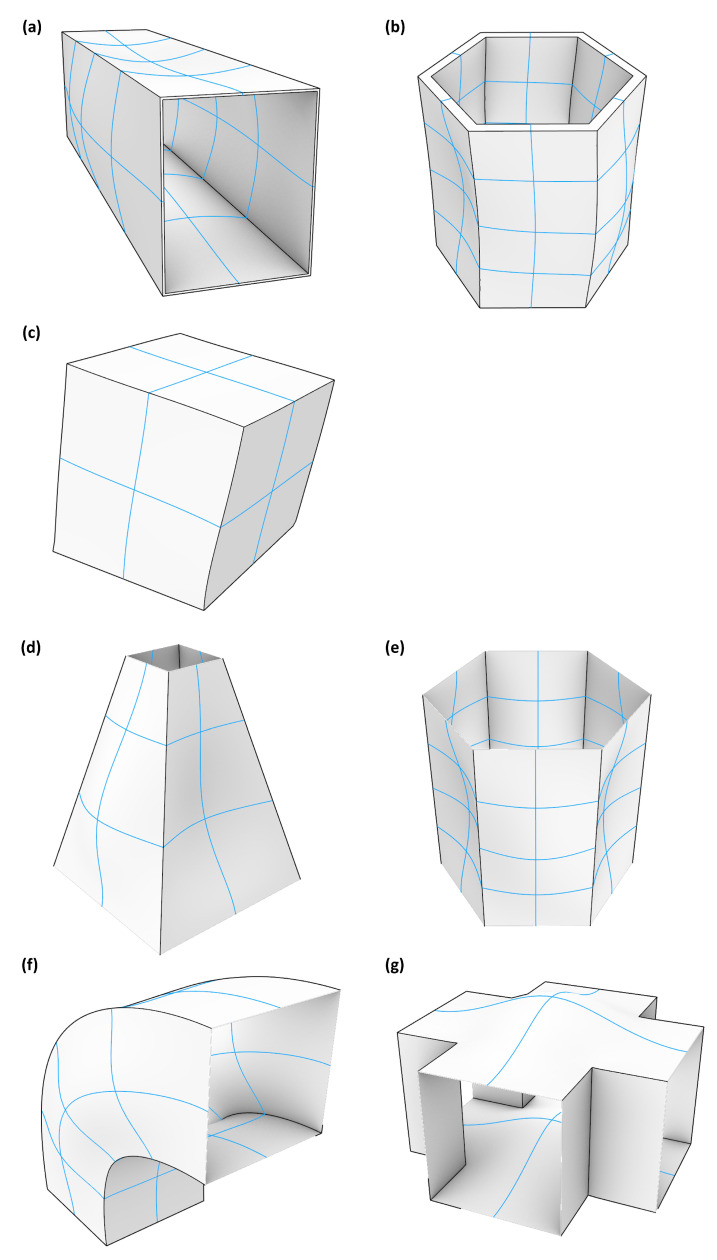
Three dimensional view of the investigated structures pre-deformed according to the 1st mode shape, including (**a**) hollow cuboid ($\delta_{\max}$ = 90 mm), (**b**) thick hexagonal prism ($\delta_{\max}$ = 15 mm), (**c**) cube ($\delta_{\max}$ = 15 mm), (**d**) truncated pyramid ($\delta_{\max}$ = 15 mm), (**e**) thin hexagonal prism ($\delta_{\max}$ = 15 mm), (**f**) curved rectangular duct ($\delta_{\max}$ = 15 mm), and (**g**) connector square ($\delta_{\max}$ = 15 mm). For visualization purposes, blue curves are imbedded in the pre-deformed surfaces to illustrate the deformations.

**Figure 13 biomimetics-09-00241-f013:**
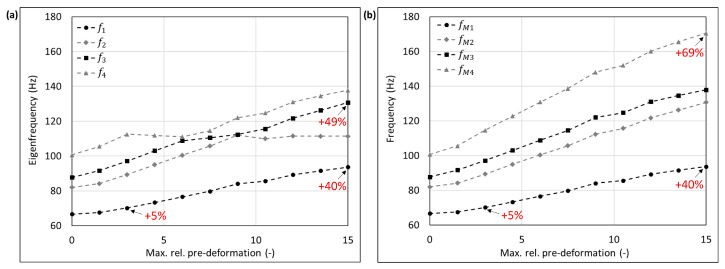
Eigenfrequencies from $f_{1}$ to $f_{2}$ (**a**) and mode shape frequencies from $f_{M1}$ to $f_{M4}$ (**b**) of the hollow cuboid pre-deformed according to mode 1, considering different maximum relative pre-deformations. For two pre-deformations, the frequency deviation compared to the undeformed structure of the mode shape adapted to the structure is given in red. In addition, the maximum obtained frequency increase is also noted.

**Figure 14 biomimetics-09-00241-f014:**
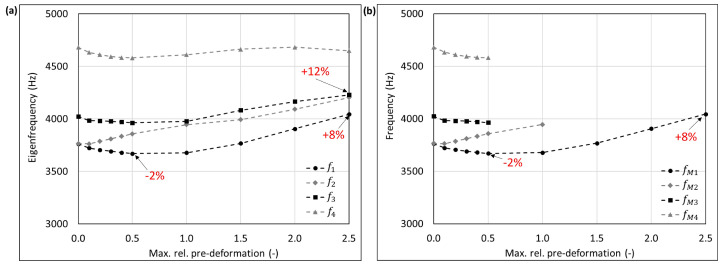
Eigenfrequencies from $f_{1}$ to $f_{2}$ (**a**) and mode shape frequencies from $f_{M1}$ to $f_{M4}$ (**b**) of the thick hexagonal prism pre-deformed according to mode 1, considering different maximum relative pre-deformations. For two pre-deformations, the frequency deviation compared to the undeformed structure of the mode shape adapted to the structure is given in red. In addition, the maximum obtained frequency increase is also noted.

**Figure 15 biomimetics-09-00241-f015:**
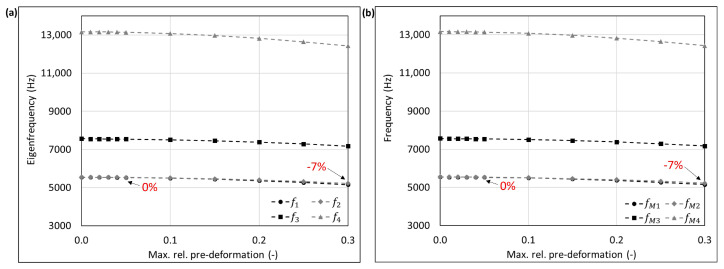
Eigenfrequencies from $f_{1}$ to $f_{2}$ (**a**) and mode shape frequencies from $f_{M1}$ to $f_{M4}$ (**b**) of the cube pre-deformed according to mode 1, considering different maximum relative pre-deformations. For two pre-deformations, the frequency deviation compared to the undeformed structure of the mode shape adapted to the structure is given in red. In addition, the maximum obtained frequency increase is also noted.

**Figure 16 biomimetics-09-00241-f016:**
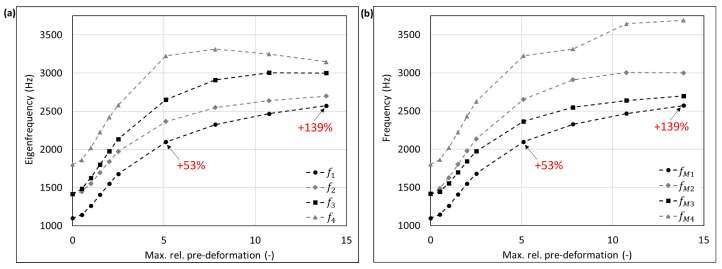
Eigenfrequencies from $f_{1}$ to $f_{2}$ (**a**) and mode shape frequencies from $f_{M1}$ to $f_{M4}$ (**b**) of the truncated pyramid pre-deformed according to mode 1, considering different maximum relative pre-deformations. For two pre-deformations, the frequency deviation compared to the undeformed structure of the mode shape adapted to the structure is given in red. In addition, the maximum obtained frequency increase is also noted.

**Figure 17 biomimetics-09-00241-f017:**
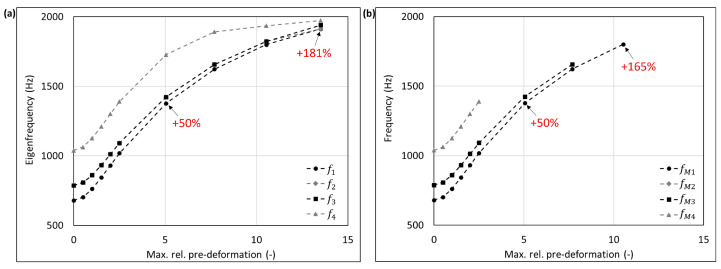
Eigenfrequencies from $f_{1}$ to $f_{2}$ (**a**) and mode shape frequencies from $f_{M1}$ to $f_{M4}$ (**b**) of the thin hexagonal prism pre-deformed according to mode 1, considering different maximum relative pre-deformations. For two pre-deformations, the frequency deviation compared to the undeformed structure of the mode shape adapted to the structure is given in red. In addition, the maximum obtained frequency increase is also noted.

**Figure 18 biomimetics-09-00241-f018:**
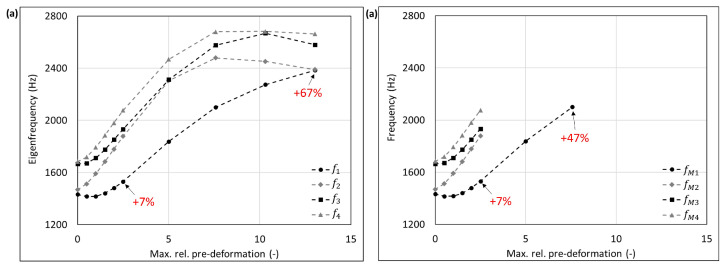
Eigenfrequencies from $f_{1}$ to $f_{2}$ (**a**) and mode shape frequencies from $f_{M1}$ to $f_{M4}$ (**b**) of the curved rectangular duct pre-deformed according to mode 1, considering different maximum relative pre-deformations. For two pre-deformations, the frequency deviation compared to the undeformed structure of the mode shape adapted to the structure is given in red. In addition, the maximum obtained frequency increase is also noted.

**Figure 19 biomimetics-09-00241-f019:**
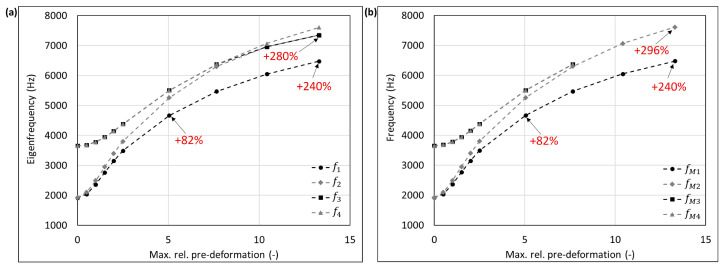
Eigenfrequencies from $f_{1}$ to $f_{2}$ (**a**) and mode shape frequencies from $f_{M1}$ to $f_{M4}$ (**b**) of the connector square pre-deformed according to mode 1, considering different maximum relative pre-deformations. For two pre-deformations, the frequency deviation compared to the undeformed structure of the mode shape adapted to the structure is given in red. In addition, the maximum obtained frequency increase is also noted.

**Figure 20 biomimetics-09-00241-f020:**
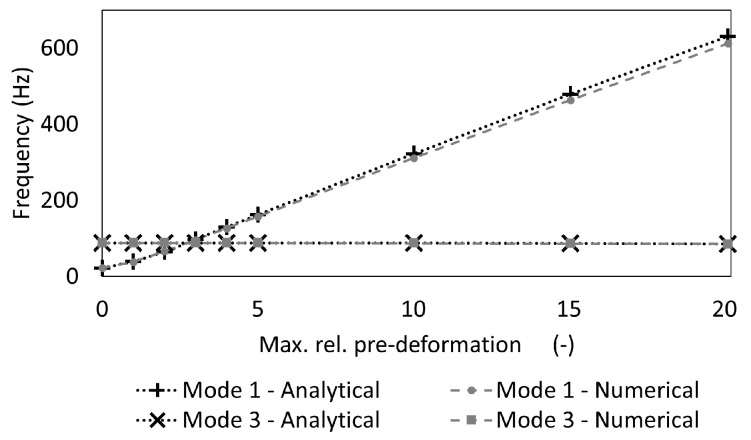
Analytically and numerically (i.e., using the created software component) obtained frequencies of the first and third mode shape of the beam pre-deformed according the first mode shape. Some data points are almost identical, which is why some markers are printed above others.

**Figure 21 biomimetics-09-00241-f021:**
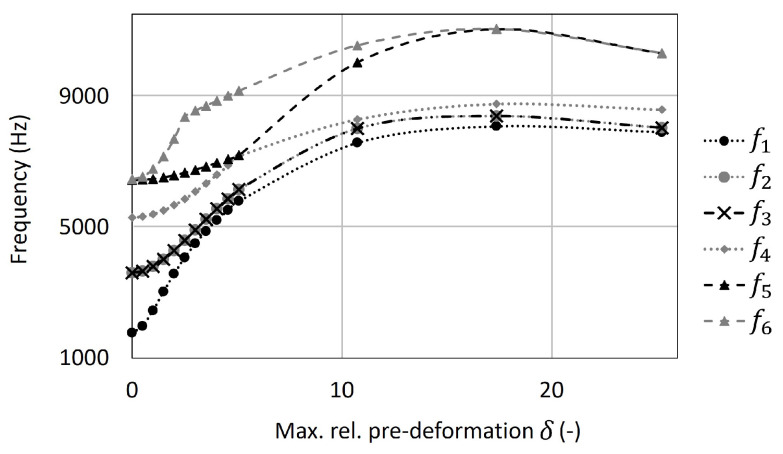
The numerical result of the first six eigenfrequencies of the square plate pre-deformed according to the first mode shape using the created shell component. Some data points are almost identical, which is why some markers are printed above others.

**Table 1 biomimetics-09-00241-t001:** Characteristics of the investigated models.

	Hollow Cuboid	Thick Hexagonal Prism	Cube	Truncated Pyramid	Thin Hexagonal Prism	Curved Rectangular Duct	Connector Square
**Mesh element type**	CTETRA (volume)	CTETRA (volume)	CTETRA (volume)	CQUAD1 (shell)	CQUAD1 (shell)	CQUAD1 (shell)	CQUAD1 (shell)
**Element edge length**	9 mm	3 mm	5 mm	4 mm	3 mm	1 mm	1 mm
**Number of elements**	1,303,473	26,040	2402	3104	13,464	2996	2370
**Material properties**	Structural steel (Young’s modulus: 210,000 MPa, density: 7850 kg m^3^, Poisson’s ratio: 0.3)						
**Studied max. pre-deformation $\delta_{\max}$**	0–150 mm (δ = 0–15)	0–25 mm (δ = 0.0–2.5)	0–30 mm (δ = 0.0–3.0)	0–25 mm (δ = 0.0–13.9)	0–25 mm (δ = 0.0–13.5)	0–25 mm (δ = 0.0–13.0)	0–25 mm (δ = 0.0–13.3)
**Wall thickness**	10 mm (constant)	10 mm (constant)	-	1.80–2.00 mm (varied)	1.85–2.00 mm (varied)	1.92–2.00 mm (varied)	1.88–2.00 mm (varied)
**Mass**	0.65–0.71 t (varied up to 9%)	8.88–9.49 kg (varied up to 7%)	7.85–8.35 kg (varied up to 6%)	0.78 t (constant)	1.88 kg (constant)	1.14 t (constant)	0.565 t (constant)

**Table 2 biomimetics-09-00241-t002:** Constant wall thickness *t* (mm) defined for the studied 3D structures truncated pyramid, thin hexagonal prism, curved rectangular duct, and connector square pre-deformed according to the 1st and 2nd mode shapes considering different maximum pre-deformations $\delta_{\max}$.

	$\delta_{\max}$ (mm)	0	1	2	3	4	5	10	15	20	25
**Adaptation** **according** **to the** **1st mode shape**	**Truncated pyramid**	2.00	2.00	2.00	2.00	2.00	2.00	1.96	1.92	1.86	1.80
	**Thin hexagonal prism**	2.00	2.00	2.00	2.00	2.00	2.00	1.98	1.95	1.90	1.85
	**Curved rectangular duct**	2.00	2.00	2.00	2.00	2.00	2.00	2.00	1.98	1.94	1.92
	**Connector square**	2.00	2.00	2.00	2.00	2.00	2.00	1.98	1.96	1.92	1.88
**Adaptation** **according** **to the** **2nd mode shape**	**Truncated pyramid**	2.00	2.00	2.00	2.00	2.00	2.00	1.96	1.92	1.86	1.80
	**Thin hexagonal prism**	2.00	2.00	2.00	2.00	2.00	2.00	1.98	1.97	1.95	1.92
	**Curved rectangular duct**	2.00	2.00	2.00	2.00	2.00	2.00	2.00	1.98	1.94	1.92
	**Connector square**	2.00	2.00	2.00	2.00	2.00	2.00	1.98	1.96	1.92	1.88

**Table 3 biomimetics-09-00241-t003:** Material and geometrical properties of the beam.

Property	Value
Young’s modulus *E* (MPa)	69,000
Material density ρ (kg m^3^)	2688
Poisson’s ratio ν (-)	0.34
Beam length *l* (m)	0.6
Beam cross-section radius *r* (m)	0.002

**Table 4 biomimetics-09-00241-t004:** Average percentage of eigenfrequency deviation between the numerical results presented in [37] and the numerical results obtained from the created shell component for different maximum relative pre-deformations δ applied to the square plate.

δ	Average Eigenfrequency Deviation $\Delta f_{\mathrm{comp}}$
1.0	0.0%
2.0	0.0%
3.0	0.1%
4.1	0.2%
5.1	0.4%
10.7	2.8%
17.4	4.7%
25.3	8.7%

**Table 5 biomimetics-09-00241-t005:** Average percentage of eigenfrequency deviation between the numerical results presented in Section 3.1 and the numerical results obtained from the created solid component for different maximum relative pre-deformations δ applied to the hollow cuboid.

δ	Average Eigenfrequency Deviation $\Delta f_{\mathrm{comp}}$
1.5	0.4%
3.0	0.3%
4.5	0.2%
6.0	0.2%
7.5	0.2%
9.0	1.4%
10.5	0.3%
12.0	1.0%
13.5	0.8%
15.0	0.7%

## Data Availability

Data are contained within the article.

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
