# Peer review of "Diatom-Inspired Structural Adaptation According to Mode Shapes: A Study on 3D Structures and Software Tools"

_biomimetics, 2024, doi:10.3390/biomimetics9040241_

Round 1

Reviewer 1 Report

Comments and Suggestions for Authors

The present work introduces valid arguments indicating that diatom frustules may be naturally optimized to withstand certain vibrational loads, likely as a strategy to resist blunt predator attacks. The authors argue that diatom frustules are responsive to certain vibrational modes and they may render them even more interesting to improve engineering designs. Taking inspiration from diatom’s structural adaptation method, computational tools were developed to study the alterations in frequency values of different 3D structures through a mode shape adaptation method. Noteworthy, the authors also provided a low-code Software version of the bio-inspired method to allow access of a wider range of non-specialized readers to the methodology. Additionally, authors included a comparison of the here obtained numerical results from the study of different 3D shapes with precedents in literature, generally reporting good correlation.

The second half of the abstract contains some confusing phrasing, linking the findings of this work and their potential applications. The abstract could benefit from a more clear and concise description of the potential real applications implicated.

In the introduction, between lines 65 and 72, the limitation of other optimization approaches, like structural optimization, is highlighted. The reader could benefit from a more detailed comparison between such traditional approaches and a bio-inspired approach.

It is not made clear which are the specific reasons for 3D-structures not being usually subjected to mode shape adaptation, and how the authors have surpassed such difficulties. Please specify these details a bit more in the main text.

The “FE” abbreviation on Figures 2 and 5 are not explained.

Lines 259-260 “increased with”.

The content of Figures 13 to 19 seems too extensive to be included in the manuscript. A more efficient alternative could be to show 2 to 3 representative examples in the main text and include the rest in the Supporting Information.

As a final remark, it could be interesting to comment on the potential usefulness of directly applying patterns or 3D structure components of real frustules into the 3D shapes studied; or else to comment on the potential applicability of this Software to study the vibrational parameters of real diatom frustules.

Comments on the Quality of English Language

Some misapproached english textes should be revised.

Reviewer 2 Report

Comments and Suggestions for Authors

The subject of the paper is on the application of structural adaptations inspired by diatoms to enhance vibrational characteristics. The subject is interesting with potential impact and applications. However, the methodology and presentation of the data have to be highly improved for the publication.

More explanation has to be provided on the existing methods on enhancing damping properties of structures and how those would have conflict with a lightweight design?

Also, the differentiation and advantage of the carried out study compared to the studies on improvement of damping properties for vibrations must be clearly described.

Comments on the Quality of English Language

Minor editing of English language required

Reviewer 3 Report

Comments and Suggestions for Authors

The title of this paper includes the phrase “Diatom-Inspired”. What is the meaning of diatom-inspired from the engineering point of view? It is not clear to the reviewer. It might to be a frustule of a diatoms and it would be an effect of the thickness of the shell in structure. However, the authors select seven elements with different features and different dimensions in this paper and then it is quite tough to the reviewer to identify what the authors evaluate from the engineering point of view in this paper. The authors should describe it in the abstract and in the introduction.

The reviewer requests the authors to modify following points in order to clarify the contents of the paper.

No.1

p.4, line 138

The authors select seven different 3D structures. The authors should describe why these geometries including sizes are selected.

No.2

p.7, figure 4

Authors describe that figure 4 shows the 1st and 2nd mode shape of the structure and color is a normalized amplitude. What kind of displacement is normalized in figure 4?

No.3

p.8 figure 7

In figure 7, the arrows go from OUTPUT to mode shape adaptation component. Is it correct?

No.4

P.10, line 201

The authors use CBEAM element in the numerical analysis. The reviewer recommends the authors to describe briefly about CBEAM element. The reviewer also recommends to describe about COUAD4 as well.

No.5

p.11 line 224

Equation 9 is written in red. Is there any reason why this equation written in red?

No.6

p.12 line 258

There is a sentence. “For the truncated pyramid (Figure 16), the thin hexagonal prism (Figure 17), the curved rectangular duct (Figure 18), and the connector square (Figure 19), the studied frequencies first increased strongly. However, with increasing maximum relative pre-deformation, the slope decreased. In figure 16 and 18, the eigenfrequency in the first mode decreases after max pre-deformation exceed about 8mm. However, that in figure 17 and 19 increases with the increase of max pre-deformation. The reviewer recommends the authors to check it.

No.7

p.12 line 259

There is a sentence. “The frequencies of the hollow cuboid (Figure 13) increase dwith a nearly constant slope for all studied pre-deformations,…”. The sentence should be .. (Figure 13) increased with a ...

No.8

p.12 line 261

There is a sentence. “ initially showed frequency decreases before the frequency values then …”. What is the frequency values in the sentence?

No.9

p.12 line 266

There is a sentence. “Mode shape switches were partly visible, i.e., the mode shape order changed due to the structural deformations.. The reviewer requests the authors to identify the figure exhibiting this feature.

No.10

p.12 line 268

There is a sentence. “the mode shapes of the curved rectangular duct and the connector square varied strongly with increasing maximum relative pre-deformation, . The reviewer thinks that the truncated pyramid (Figure 16) also shows similar feature.

No.11

p.14, figure 12

In figure 12, truncated pyramid and thin hexagonal prism are shown in (d) and (e), while those are in (e) and (d) in figure 3. The reviewer recommends the authors to keep order of figures.

No. 12

p.15 Figure 13

The authors should describe the name of parameters, f1, f2, f3, f4, M1, M2, M3 and M4, respectively.

No. 13

p. 16 Figure 14

Some plots in the figure terminated at the lower Max rel. pre-deformation (e.g. M4). Similar feature is also observed in other figures. The reviewer requests the authors to describe the possible reasons.

No. 14

p.21 Figure 20

The reviewer thinks that it is difficult to distinguish each plot. The reviewer requests the authors to change the marker and to add the explanation. The plot, f2, in figure 21 is also tough to distinguish.

No. 15

p.22 line 311

There is a sentence. “… with the exception of the 2nd and 3rd eigenfrequencies, which exhibited a decrease at δ = 10.7, . It seems to the reviewer that the third eigenfrequency does not decrease atδ = 10.7, In addition, it is difficult to identify the second eigenfrequency in the figure.

Round 2

Reviewer 2 Report

Comments and Suggestions for Authors

The authors did a minor revision.
No further explanation was provided about the existing methods on enhancing damping properties of structures. It is still not clear how those methods would have conflict with a lightweight design.

The novelty and advantage of the carried out study compared to the studies on improvement of the damping properties for vibrations are well explained.

Comments on the Quality of English Language

Minor editing of English language required.

Author Response

Dear reviewer!

We thank you very much for reviewing our revised manuscript and your critical and helpful comments on our work! Below, we respond to your comment and report the changes on the manuscript that we conducted. Thanks to your suggestion we were able to improve our manuscript and make it more comprehensible. We also want to thank you for again checking the revised manuscript.

Kind regards on behalf of all authors,

Simone Andresen

Comment:

“The authors did a minor revision.
No further explanation was provided about the existing methods on enhancing damping properties of structures. It is still not clear how those methods would have conflict with a lightweight design.

The novelty and advantage of the carried out study compared to the studies on improvement of the damping properties for vibrations are well explained.”

Response to the general comment:

Thank you for this comment. We agree with you that we did not include detailed information about methods to increase damping in lightweight structures.

We did a literature review on this topic, summarized the information and included it to the manuscript text (page 2, lines 60-82).

Reviewer 3 Report

Comments and Suggestions for Authors

The reviewer thinks that the authors copprespond to the reviewer's comments. 

Author Response

Dear reviewer,

thank you very much again for taking your time to review our manuscript and for all your suggestions and comments. We are looking forward to the publication of the now improved manuscript.

Thanks again and kind regards,

Simone Andresen